# The Evolution of Highly Pathogenic Avian Influenza A (H5) in Poultry in Nigeria, 2021–2022

**DOI:** 10.3390/v15061387

**Published:** 2023-06-17

**Authors:** Clement Meseko, Adelaide Milani, Bitrus Inuwa, Chinonyerem Chinyere, Ismaila Shittu, James Ahmed, Edoardo Giussani, Elisa Palumbo, Bianca Zecchin, Francesco Bonfante, Silvia Maniero, Angélique Angot, Mamadou Niang, Alice Fusaro, Federica Gobbo, Calogero Terregino, Taiwo Olasoju, Isabella Monne, Maryam Muhammad

**Affiliations:** 1Regional Laboratory for Animal Influenza & Transboundary Diseases, National Veterinary Research Institute (NVRI), Vom 930101, Nigeria; usfilmalgwi@yahoo.com (B.I.); chinoscnc@gmail.com (C.C.); ismaila.shittu@gmail.com (I.S.); etsujamesone@gmail.com (J.A.); maryam.muhammad@nvri.gov.ng (M.M.); 2Istituto Zooprofilattico Sperimentale delle Venezie (IZSVe), Division of Comparative Biomedical Sciences (BSBIO), 35128 Padova, Italy; amilani@izsvenezie.it (A.M.); egiussani@izsvenezie.it (E.G.); epalumbo@izsvenezie.it (E.P.); bzecchin@izsvenezie.it (B.Z.); fbonfante@izsvenezie.it (F.B.); smaniero@izsvenezie.it (S.M.); afusaro@izsvenezie.it (A.F.); fgobbo@izsvenezie.it (F.G.); cterregino@izsvenezie.it (C.T.); imonne@izsvenezie.it (I.M.); 3Animal Health Service (NSAH), Food and Agriculture Organization of the United Nations (FAO-UN), 00198 Rome, Italy; angelique.angot@fao.org; 4Regional Office for Africa (RAF), Emergency Centre for Transboundary Animal Diseases (ECTAD), Food and Agriculture Organization of the United Nations (FAO-UN), Accra 00233, Ghana; mamadou.niang@fao.org; 5Federal Department of Veterinary and Pest Control Services (FDV&PCS), Federal Ministry of Agriculture and Rural Development (FMARD), Abuja 900108, Nigeria; 7taiwo@gmail.com

**Keywords:** highly pathogenic avian influenza, H5N1/H9N2 reassortant virus, Nigeria

## Abstract

In 2021, amidst the COVID-19 pandemic and global food insecurity, the Nigerian poultry sector was exposed to the highly pathogenic avian influenza (HPAI) virus and its economic challenges. Between 2021 and 2022, HPAI caused 467 outbreaks reported in 31 of the 37 administrative regions in Nigeria. In this study, we characterized the genomes of 97 influenza A viruses of the subtypes H5N1, H5N2, and H5N8, which were identified in different agro-ecological zones and farms during the 2021–2022 epidemic. The phylogenetic analysis of the HA genes showed a widespread distribution of the H5Nx clade 2.3.4.4b and similarity with the HPAI H5Nx viruses that have been detected in Europe since late 2020. The topology of the phylogenetic trees indicated the occurrence of several independent introductions of the virus into the country, followed by a regional evolution of the virus that was most probably linked to its persistent circulation in West African territories. Additional evidence of the evolutionary potential of the HPAI viruses circulating in this region is the identification in this study of a putative H5N1/H9N2 reassortant virus in a mixed-species commercial poultry farm. Our data confirm Nigeria as a crucial hotspot for HPAI virus introduction from the Eurasian territories and reveal a dynamic pattern of avian influenza virus evolution within the Nigerian poultry population.

## 1. Introduction

Poultry operations in Nigeria faced many challenges amidst the Coronavirus disease (COVID-19) pandemic, including global food insecurity and economic downturn [1]. Though the problems associated with highly pathogenic avian influenza (HPAI) in Nigeria and many countries predate the COVID-19 pandemic, persistent outbreaks, morbidity, and mortality in chickens and other birds have continued unabated and worsened the economic and public health impacts of these problems. The fear that HPAI, a zoonotic disease, could be a progenitor of another pandemic is rife and driven by ecological, behavioural, and socioeconomic changes, as well as the ability of this virus to rapidly evolve through both genetic reassortment and the acquisition of mutations [2,3].

The HPAI phenotype is caused by the H5 and H7 subtypes of the influenza A virus (IAV) of the Orthomyxoviridae family; this virus occurs naturally in wild waterfowls as a low pathogenic avian influenza (LPAI) virus and can mutate into HPAI [4] when transmitted to poultry. Influenza A viruses (IAV) are divided into subtypes based on their combinations of the two surface glycoproteins HA and NA; in birds, 16 HA and 9 NA types have been identified [5].

Over the last two decades, the world has witnessed multiple intercontinental epidemic waves of the H5Nx subtype of HPAI from Asia and Europe to Africa [6] as a consequence of the spread of viruses descended from the H5N1 virus A/goose/Guangdong/1/1996 (Gs/GD), which was first detected in China in 1996. Since their emergence, these viruses have dramatically expanded their geographical distribution, spreading to distinct continents and remaining well entrenched in a number of countries. They have been acquiring a broader host range, with a constantly expanding list of free-living wild bird species, which can potentially be affected by the virus, with spill-over events in mammals being increasingly reported [7,8].

Since 2006, multiple Gs/GD/96 H5Nx lineages have been introduced in Nigeria, which have had a negative impact on poultry productivity and food security and threatened public health. The first introduction of the Gs/Gd lineage of the HPAI clade 2.2 to Nigeria happened during the harmattan (winter) season of 2006 [9,10]. Between 2006 and 2008, when the epidemic occurred, millions of poultry birds either died or were culled to contain the further spread of the disease and the economic losses of farmers [11]. These control efforts paid off and the epizootic was declared over by a self-declaration of disease-free status at the World Organization for Animal Health (WOAH, formerly OIE) General Assembly in 2013 (https://www.woah.org/app/uploads/2022/06/eng-archive-2000-may-20222.pdf, accessed on 15 March 2023) [12].

Not long afterwards, a more devastating resurgence of another strain of HPAI caused by clade 2.3.2.1c was witnessed in 2015/2016 [13,14]. Thereafter, a series of re-introductions of various subtypes of HPAI into the Nigerian agro-ecological space, including the subtype H5N8, became more frequent [15]. H5N6 was detected in live bird markets (LBMs) in 2019 [16]. Adding to this plethora of outbreaks and detections of H5Nx in Nigeria, the low pathogenic avian influenza (LPAI) subtype, H9N2, has been added to the mix and is frequently and widely isolated in poultry, especially at LBMs [17].

Apparently, Nigeria has emerged as a regional hotspot of HPAI in sub-Saharan Africa [6] and accounts for a significant proportion of the cases reported to the WOAH from the African region over the last two decades (https://www.woah.org/en/disease/avian-influenza/#ui-id-2, accessed on 15 March 2023). More intriguing is the diversity of HPAI and LPAI that has been detected in the Nigerian agro-ecological space since 2006. HPAI of the H5N2 subtype has previously been detected in healthy wild waterfowl in Nigeria [18]. By 2011, Snoeck et al. [19] again reported the detection of reassortant LPAI H5N2 viruses in African wild birds sampled in the Hadejia-Nguru wetlands of northern Nigeria. Similarly, in 2013, Coker et al. [12] recovered genetically similar strains of H5N2 in commercial waterfowls at an LBM in southwestern Nigeria. These previous H5N2 strains were limited in their spread, as subsequent surveillance activities did not detect the H5N2, even in LBMs [16,17].

The agro-ecological source of the introduction of AIV into Nigeria is not fully understood and has been poorly investigated [10]. We can merely theorize based on the available epidemiological and genetic data that AIV is most likely introduced seasonally to Nigeria through activities and contact with migratory waterfowls from Eurasia [20]. Nigeria remains a country at risk of the continuous new introduction of AIV, with chances that an endemic situation may result in the co-circulation, co-infection, and emergence of possible new reassortant viruses.

Since early January 2021, new outbreaks of HPAI H5Nx have caused multiple outbreaks across the country and been reported in 30 out of the 37 administrative territories (Pers Comm: Federal Department of Veterinary and Pest Control Services), resulting in 467 positive cases as of December 2022. The transmission of these new HPAIVs in poultry flocks, where the H9N2 subtype is also entrenched in the poultry population, is a matter of great concern given the proclivity for the reassortment of AIV [17]. Using a wide collection of samples gathered throughout the 2021 and 2022 epidemic events in Nigeria, we investigated the genetic diversity of the AIVs circulating across the Nigerian states and provided evidence of the emergence of a putative H5N1/H9N2 reassortant in a mixed-species commercial poultry farm.

## 2. Materials and Methods

### 2.1. Sample Collection and Necropsy

Diagnostic outbreak investigations were carried out following a suspicion of HPAI outbreaks in Nigeria. From previous experience, poultry farmers are sensitive to abnormal mortality patterns and are quick to raise the index of suspicion by contacting the nearest private and state veterinarians. The attending veterinarians’ initial observations, with respect to clinical manifestations, were documented. Fresh carcasses were selected and shipped under cold-chain to the Regional Laboratory for Animal Influenza, National Veterinary Research Institute (NVRI) in Vom, for diagnosis and necropsy. Parenchymatous organs and tissues were harvested from the carcasses, pooling several tissues from the same flocks and species for further processing.

### 2.2. Virus Identification and Isolation

The issues were homogenized and supernatant fluid was collected for a nucleic acid extraction using the Qiagen Viral RNA kit (Qiagen, Hilden, Germany), following the manufacturer’s instructions. The total RNA was extracted from a pool of combined tissues and first screened for the IAV matrix gene (M-gene) and H5, H7, H9, and Nx IAV subtypes via an RT-qPCR using the QuantiTect Multiplex (Qiagen, Hilden, Germany) and OneStep RT-PCR (Qiagen, Hilden, Germany) kits, as previously described [21,22,23,24,25]. Influenza-A-positive samples were isolated and typed according to the OIE Terrestrial Manual (https://www.woah.org/fileadmin/Home/eng/Animal_Health_in_the_World/docs/pdf/2.03.04_AI.pdf, accessed on 15 March 2023).

To ascertain the presence of an H5N2 reassortant virus in the samples potentially co-infected with H9N2 and HPAI H5N1 viruses, homogenate supernatants were subjected in parallel to cloning through multiple rounds of limiting dilution in embryonated chicken eggs and plaque purification in MDCK cells. Briefly, for the limiting dilution cloning in the eggs, the original sample was serially diluted in a phosphate-buffered solution with antibiotics and incubated for 2 h at room temperature with chicken polyclonal antiserum raised against a G1 lineage H9N2 virus before its inoculation in the eggs. To plaque purify the virus, the homogenate supernatant was serially diluted in cell culture medium and plated onto the MDCK cells for 1 h at 37 °C in 5% CO2. After the removal of the inoculum, a 0.9% agar overlay was added and the plates were incubated for 72 h. The allantoic fluid and several plaques were collected from the eggs/cells infected with the highest dilution and underwent further diagnostic screening and NGS analyses.

### 2.3. Genomic Sequencing and Phylogenetic Analysis

Ninety-seven HPAI-H5-positive clinical specimens were selected based on their epidemiological locations, species, and time of outbreak and sent to the EURL, WOAH, and FAO reference laboratories for avian influenza (AI) in Italy (Istituto Zooprofilattico Sperimentale delle Venezie), where they underwent sequencing and genetic data analyses.

A target RT-PCR approach was used to amplify the influenza A virus whole genomes, as previously described [6]. The sequencing libraries were obtained using the Nextera XT DNA Sample preparation kit (Illumina, San Diego, CA, USA) and sequenced on an MiSeq instrument (San Diego, CA, USA) using the 2 × 250 bp PairedEnd [PE] mode.

The raw sequencing reads produced by the MiSeq instrument were cleaned with Trimmomatic v0.32 [26] with a minimum quality of 20. Illumina Nextera XT adapter sequences were clipped from the reads using scythe v0.991 (https://github.com/vsbuffalo/scythe, accessed on 15 March 2023) and amplification primers were removed using sickle v1.33 (https://github.com/najoshi/sickle, accessed on 15 March 2023). Reads that were unpaired or shorter than 80 bp were discarded. The cleaned reads were aligned against a reference genome using the MEM algorithm from BWA v0.7.1229. Picard tools v2.1.0 (http://broadinstitute.github.io/picard/, accessed on 15 March 2023) and GATK v3.530-32 [27] were used to improve the alignment quality, correct potential errors, and recalibrate the base quality score. LoFreq v2.1.233 [28] was used to call the single-nucleotide polymorphisms (SNPs).

The consensus sequence was created with an in-house script. The consensus sequences were aligned using the MAFFT v7 online server (https://mafft.cbrc.jp/alignment/server/, accessed on 15 March 2023) [29] and compared with the sequences of the most related virus strains available in GISAID. Maximum likelihood phylogenetic trees of each gene segment were obtained using IQTREE v1.6.6 (https://github.com/iqtree/iqtree1, accessed on 15 March 2023) and their robustness was determined via an ultrafast bootstrap resampling analysis of 1000 replications [30,31]. The phylogenetic trees were visualized using the FigTree v1.4.4 software (http://tree.bio.ed.ac.uk/software/figtree/, accessed on 15 March 2023).

The FluSurver tool (https://flusurver.bii.a-star.edu.sg/, accessed on 15 March 2023) was used to identify the amino acid mutations in the influenza A virus proteins, which could affect biological functions such as virulence, host adaptation, and drug resistance.

## 3. Results

### 3.1. Virus Identification and Pathological Findings

Depression, somnolence, drooling fluid from the mouth, diarrhoea, hock sitting, inappetence, a drop in egg production, and a sudden/high mortality were the most frequent clinical signs raising the suspicion of an HPAI infection. The laboratory investigations applied to the samples collected from the farms reporting HPAI suspicions allowed for the confirmation of 467 outbreaks of HPAI H5Nx across 31 administrative regions in Nigeria from January 2021 to December 2022 (Figure 1).

More specifically, 454 A(H5N1), 12 A(H5N8), and 1 A(H5N2) outbreaks were reported from the total of the 467 outbreaks reported in the country. It is of note that the unique A (H5N2) subtype was identified in a tissue sample collected from a mixed-species poultry flock of 20,792 birds (8271 ducks and 12,521 layer chickens) in Toro LGA of Bauchi State, northeastern Nigeria (Longitude:10.056813 and Latitude:09.0685465) (Figure 1), where a sudden increase in mortality was registered (2000 birds within 24 h).

The HPAI H5N1 and H5N8 outbreaks were reported in a mixture of backyard, semi-intensive, and intensive commercial operations with mixed species, mainly including chickens (broiler, layers, and growers) and ducks (Appendix A). Necropsy revealed gross lesions that were typical of HPAI, including diffuse and ecchymotic haemorrhages, which are evidence of capillary damage. The congestion observed in the carcasses of commercial chickens was subtle and did not involve the multiple organ damage that is known to accompany HPAI. The gross lesions observed among the peacocks, geese, and broilers/indigenous chickens were classical and included cyanoses of combs, beaks, and wattles, subcutaneous haemorrhages on the shank, hock joints, and breast muscles, and ecchymotic haemorrhages in the proventriculus and ventriculus. In addition, there was hepatic congestion with a friable texture and streaks of peripheral pallor, petechial haemorrhages in the thigh and breast muscles, enlarged and congested spleens, severe peritonitis and the adhesion of visceral organs, haemorrhagic enteritis, and severe haemorrhagic tracheitis, as well as haemorrhages in the ceca and cecal tonsils. These major pathological presentations are shown in Figure 2.

### 3.2. Phylogenetic Analyses

The whole genome sequences of ninety viruses (eleven H5N8, seventy-eight H5N1, and one H5N2 virus) and partial genomes of five H5N1 and two H5N8 viruses were obtained (Appendix A).

The phylogenetic analysis of the HA genes confirmed that all the HPAI H5N1, H5N2, and H5N8 Nigerian viruses belonged to clade 2.3.4.4b. The HPAI H5N1 viruses belonged to the H5N1 A/Eurasian_Wigeon/Netherlands/1/2020-like genotype, while the HPAI H5N8 viruses belonged to the H5N8 A/duck/Chelyabinsk/1207-1/2020-like genotype. [32] The H5N2 virus is a new reassortant virus that has never been identified in Europe. An in-depth analysis of the HA phylogeny revealed the existence of two major genetic groups in Nigeria, namely 1 and 2 (Figure 3). Group 1 comprised the H5N1 (*n* = 83) and H5N2 (*n* = 1) Nigerian strains, as well as the H5N1 viruses collected in Europe (2020–2021) and other West African countries (2020–2022). In particular, the Nigerian H5N1 viruses formed different genetic clusters within Group 1, suggesting the occurrence of multiple virus introductions at the beginning of 2021, followed by a local spread. Only one of these clusters (Group 1-A), which included most of the characterized H5N1 viruses (58/83), persistently circulated in the country up to 2022. Group 2 comprised only Nigerian viruses of the H5N8 subtype, which showed the highest identity with the H5N8 viruses identified in central Asia and Europe between October 2020 and January 2021, suggesting separate virus incursions, likely through different routes. It is also of note that the H5N8 subtype was detected in the country only for few months after its introduction at the beginning of 2021 and has not been seen extensively circulating in other West African countries.

The tree topology of N1 (Appendix A), N8 (Appendix A), and the other six internal gene segments confirms a genetic clustering similar to that shown for the haemagglutinin gene for the H5N1 and H5N8 viruses, respectively. Differently, the complete genome analyses of the H5N2 virus indicated that it emerged from reassortment events between the H5N1 and low pathogenic H9N2 viruses circulating in the poultry in Nigeria. Specifically, six genes (PB2, PB1, PA, HA, NP, and NS) are highly related to the Nigerian H5N1 viruses, while two genes (NA and M) show the highest identity with the H9N2 viruses of G1 lineage identified in Nigeria between 2019 and 2021 (Figure 1, Appendix A) [17]. As for the endemic circulation of the H9N2 subtype in Nigeria and weak positivity detected by the real-time RT-PCR for the H9 subtype [25], an ultradeep sequencing approach was applied to characterize the population diversity within the H5N2-positive sample and exclude an H5N1/H9N2 coinfection. No reads ascribable to the N1 subtype were identified, suggesting the existence of an H5N2/H9N2 coinfection. Therefore, clonal purification by limiting the dilution in the embryonated eggs and MDCK cells of the isolate obtained from the original clinical material was performed and the clone was genetically characterized. Genetic analyses of the eight gene segments confirmed the presence of an H5N2 reassortant subtype.

### 3.3. Genetic Characterization

The molecular assessment of the consensus sequences of all the Nigerian viruses revealed the presence of several mutations that are likely associated with an increased zoonotic potential or adaptation to poultry. A detailed analysis of the HA gene showed that, in all the viruses except one (A/Avian/Nigeria/VRD-21-019_21RS744-61/2021), the mutation S137A (H3 numbering) and mutation S158N occurred, whereas the mutation T160A, which causes the loss of glycosylation at site 158, was present in all the H5N8 strains and in eighteen H5N1 viruses. All these mutations have been demonstrated to cause increased alpha2,6-SA binding [33,34] and, in particular substitutions, T160A can contribute to acquiring a dual receptor-binding property [35]. Glycosylation on hemagglutinin proteins needs particular attention, considering its role in the viral properties involved in virus attachment to target cell receptors, virulence, receptor-binding specificity, and host adaptation [36].

For the NA protein, most of the H5N1 Nigerian viruses (seventy-three out of eighty-two) presented a twenty-two amino acid deletion in the NA stalk region. It has been shown that this feature can increase the virulence of a virus [37] and is a marker of virus adaptation from wild aquatic birds to poultry [38,39,40].

The investigation of the polymerase complex protein revealed that the PB2 protein of the A/chicken/Nigeria/743A_22VIR3286-80/2021 virus possessed the mutation D701N, which has been recognized as an important mammalian adaptive marker related to increased replication and virulence in mammals [41,42,43,44,45]. Furthermore, the H5N1 virus A/Avian/Nigeria/271PT_22VIR3286-71/2021 presented K482R, which has been reported to increase the epolymerase activity in mammalian cell line [46].

Eighty-two out of eighty-three viruses showed the Y52H mutation in their NP protein, which seems to confer a resistance to the butyrophilin subfamily 3 member A3 (BTN3A3), a potent avian influenza virus inhibitor in humans [47]. All the viruses investigated in our study presented N30D and T215A mutations in the Matrix protein (M1), which have been proven to affect the pathogenicity of H5N1 viruses in mammals [48,49]. Mutations detected in the Matrix-2 (M2) protein and associated with amantadine resistance have been found in three viruses; in detail, the S31N mutation [50,51] in H5N2 A/chicken/Nigeria/743A_22VIR3286-800/2021 and V27I [52] in the H5N1 A/Avian/Nigeria/741_22VIR3286-29/2021 and A/Chicken/Nigeria/751_22VIR3286-35/2022 viruses.

All the viruses showed the P42S mutation in the NS1 protein, which has been described as increasing virulence in mice and also demonstrated to be critical for the H5N1 virus to antagonize the interferon induction [53]. Furthermore, the L103F and I106M mutations, which can increase virulence in mice, were detected in all the viruses analysed here [54].

All the H5N1 and H5N2 viruses showed the N205S mutation in NS1 and T48A mutation in NS2, which cause decreased antiviral responses in hosts and correspond to N200S and T47A in the study previously published by Imai et al. [55] (Table 1). Further mutations supposed to be host specificity markers according to statistical inferences are shown in Table 1.

## 4. Discussion

Nigeria has been identified as a regional hotspot for HPAI since 2006 [6,9]. How the maiden outbreak was introduced into the agro-ecological space is debatable, but its introduction through migratory waterfowls has been widely claimed [6,8,14].

In the current study, the clade 2.3.4.4b H5N1, H5N2, and H5N8 AIVs were confirmed in commercial poultry farms across Nigeria, where there is a concurrent endemic circulation of H9N2. Longitudinal outbreak investigations over the last decade have revealed the frequent seeding and reassortments of HPAI H5Nx from external sources to poultry in Nigeria [14].

In the past, the AIV subtype H5N2 has been detected in Nigeria under different agro-ecological situations and conditions, but none of these involved chickens in commercial poultry farms [12,18,19]. A molecular and phylogenetic analysis of these earlier H5N2 strains that were detected in Nigeria showed some degree of relationship, especially in their ecology and origin in relation to waterfowls from Europe [20]. None of these viral strains have become established in local poultry or other bird populations, despite the sustained active and passive surveillance for AIV in Nigeria [17].

In the last 10 years, other subtypes and clades of AIV, including the H5N1, H5N8, H5N6, and H9N2 subtypes detected in Nigeria, have mostly been the consequence of direct introduction through migratory waterfowls, except for the H9N2 subtype, which was most probably introduced through the poultry trade [6]. However, in the present study, apart from re-introductions of H5Nx at different times, there appears to be evidence of a genetic reassortment between the circulating H5N1 and H9N2 viruses. This is not surprising, as Asian countries, as well as Egypt, have had experiences similar to Nigeria, where the introduction and spread of HPAI H5Nx resulted in an inter-subtype reassortment with H9N2 viruses endemically circulating in local poultry [60,61]. However, the identification of a new reassortant H5N1/H9N2 avian influenza virus in Nigeria suggests that the emergence of reassortant highly pathogenic H5N1/H9N2 avian influenza viruses has recently become a more frequent event in West Africa. Indeed, another H5N1/H9N2 virus containing the H9N2 PA gene was reported in Burkina Faso in late 2021 [62].

Genomic and virological evidence suggests that the reassortment and emergence of the H5N2 virus in Nigeria may have taken place in Nigerian agro-ecology. Indeed, the combination of H5N1 and H9N2, where the HA and NA segments are donated by the circulating H5N1 and H9N2 parent viruses to the new progeny, appears to be unique to Nigeria. An interesting dimension of this reassortment event is also the co-existence of an entire H9N2 subtype in the same cocktail of the sample where the reassortment genes were detected, providing further proof that the N2 was donated from the H9N2 progenitor and that the emergence of the reassortant strain was most probably identified at its source. As no further outbreaks caused by this reassortant strain have been identified in the region so far, it may be suggested that the early detection of this virus prevented its further spread. However, additional experimental studies are needed to exclude that this lack of viral dissemination was the consequence of a reduced viral fitness of the emerged H5N2 virus. This also lends credence to the necessity of implementing active surveillance and virus searches in the region in order to detect these and other circulating IAV subtypes for improved control of the disease.

Analyses of the phylogenetic topologies revealed that the sequences found in Nigeria were dispersed throughout the trees, forming multiple genetic clusters, indicating the occurrence of several independent introductions of the virus into the country that occurred in previous epidemic waves [6,15]. Two main clusters can be identified, both finding their potential precursors in viruses identified in Europe and central Asia: a cluster provisionally named here as Group 1 includes most of the Nigerian H5N1 viruses and viruses circulating in Niger, Benin, Mali, Burkina Faso, and Ghana from June 2021 to January 2022, and one genetic group formed by all the Nigerian H5N8 viruses (Group 2) characterized in this study. The high genetic similarity existing between the Nigerian viruses identified within Groups 1 and 2, respectively, indicates that lateral transmission events most likely happened, determining a high number of secondary cases in the poultry sector after the virus introduction. Despite the sources of this introduction being far from conclusively identified, the phylogenies indicate that the country has been experiencing the co-circulation of viruses, most probably newly introduced from Europe through wild birds, since late 2020. Since then, these viruses have been then evolving in Western African areas. Indeed, what is distinct from previous epidemic seasons is that most of the 2021–2022 H5N1 viruses characterized here are highly related to the HPAI H5N1 viruses identified in West Africa since early 2021, suggesting a persistent circulation of the viruses in the region.

The whole genome analyses revealed the presence of molecular mutations in the HPAI H5N1 and H5N2 viruses, which could be associated with an increased zoonotic potential. Particularly interesting was the presence of the mammalian adaptive marker D701N found in the PB2 protein of the H5N2 virus A/chicken/Nigeria/743A_22VIR3286-80/2021; it was experimentally shown by Yang et al. [63] that, in a mouse-adapted H9N2 virus, this mutation conferred a higher virulence and more efficient viral replication in mouse lungs and livers. Furthermore, the H5N1 virus A/avian/Nigeria/271PT_22VIR3286-71/2021 isolated in Kano presents K482R, which has been reported to increase the polymerase activity in mammalian cell lines [46]. All the H5N1s described here present the S137A mutation (H3 numbering) (position 149 from the initial methionine), which has been demonstrated to cause increased alpha2,6-SA binding and has been already described as characterizing the HA gene of all the H5 viruses identified in Europe since early 2021 [7,33]. Furthermore, the deglycosylation at site 158N in the HA protein, caused by the substitution of T160A, can contribute to acquiring a dual receptor-binding property and could represent an important evolutionary molecular marker within clade 2.3.4.4 H5Nx [35].

## 5. Conclusions

This study confirms Nigeria as a hotspot for the virus introduction of HPAI H5Nx viruses. Despite the lack of efficient surveillance plans for wild birds across the western African region, the results of our investigations suggest that HPAI H5Nx viruses were introduced from Eurasian territories, most likely via migratory birds. The occurrence of lateral spread transmission events between distinct poultry farms indicates a biosecurity breach in the poultry system in the country, although a regional spread through wild bird movements cannot be excluded. The relatedness between the Nigerian viruses and those identified in other western African countries suggests the possible existence of inter-border livestock movement and poultry trade in these territories.

The emergence of a further putative reassortant virus in West Africa, with the presence of a mutation with zoonotic potential (D701N), added to the identification of other H5N1/H9N2 reassortants in Burkina Faso during late 2021 [62], raising concerns about the risk that western African countries may be a new source of emerging avian influenza viruses with increased zoonotic potential. Indeed, the role of the H9N2 subtype as a main donor of viral genes that result in zoonotic infections is well documented [64]. The diversified biosecurity measures applied across the country, the sales of poultry through poorly regulated live markets, and the limited financial resources available to implement the appropriate HPAI virus surveillance and control measures are all features that make the eradication of the diseases in this area an extremely challenging goal, besides increasing the chance for the virus to evolve as expected. Given the pattern of introduction and spread of IAV in Nigeria, the importance of the pre-emptive bio-surveillance of wild migratory birds in major wetlands and live bird markets, combined with stringent biosecurity measures, cannot be overemphasised. Future discussion on additional control measure such as vaccination may be an appropriate tool in some circumstances, but their use may sometimes have negative impacts. Therefore, the government must first carry out risk/benefit assessments of the situation. The economic and public health threat of the endemicity of AIV in Nigeria, as portended in this study, may have far-reaching consequences for sub-Saharan Africa and the rest of the world.

## Figures and Tables

**Figure 1 viruses-15-01387-f001:**
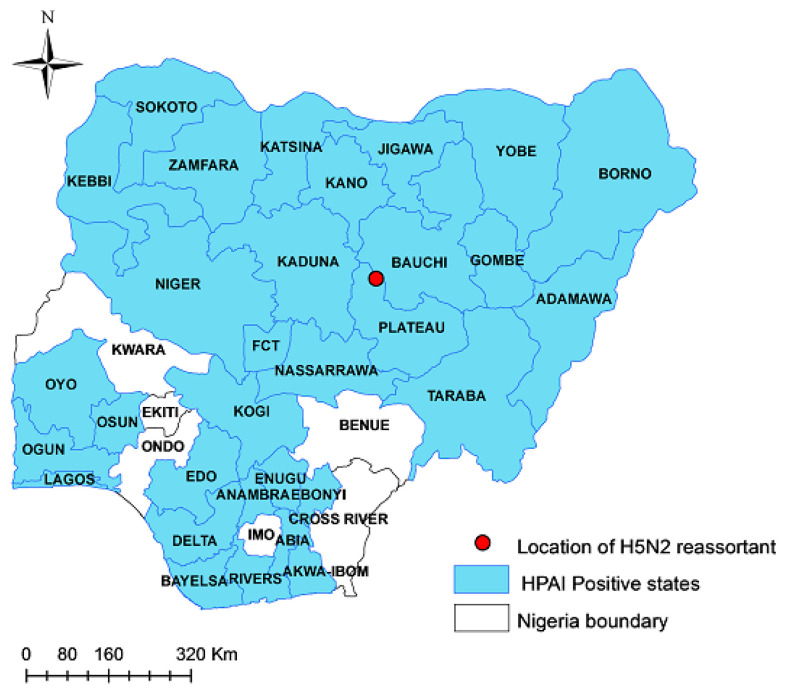
Map of Nigeria showing location and distribution of HPAI in Nigeria, 2021–2022.

**Figure 2 viruses-15-01387-f002:**
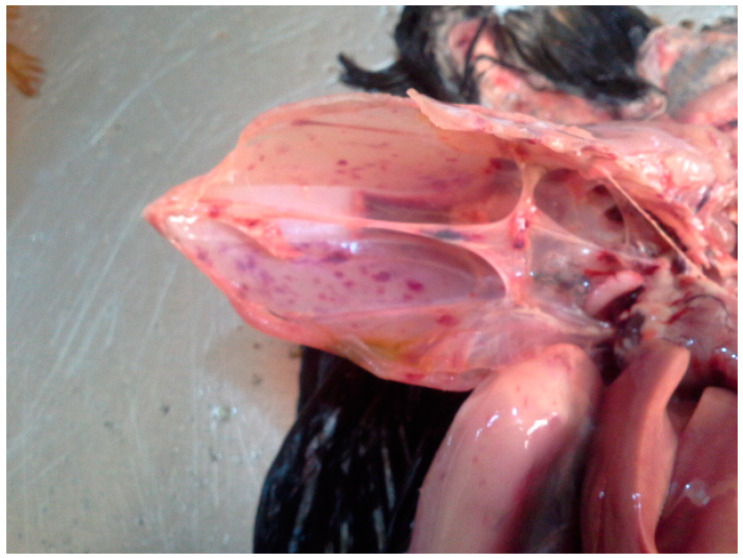
Haemorrhages of organs and tissues caused by HPAI in Nigeria, 2021–2022.

**Figure 3 viruses-15-01387-f003:**
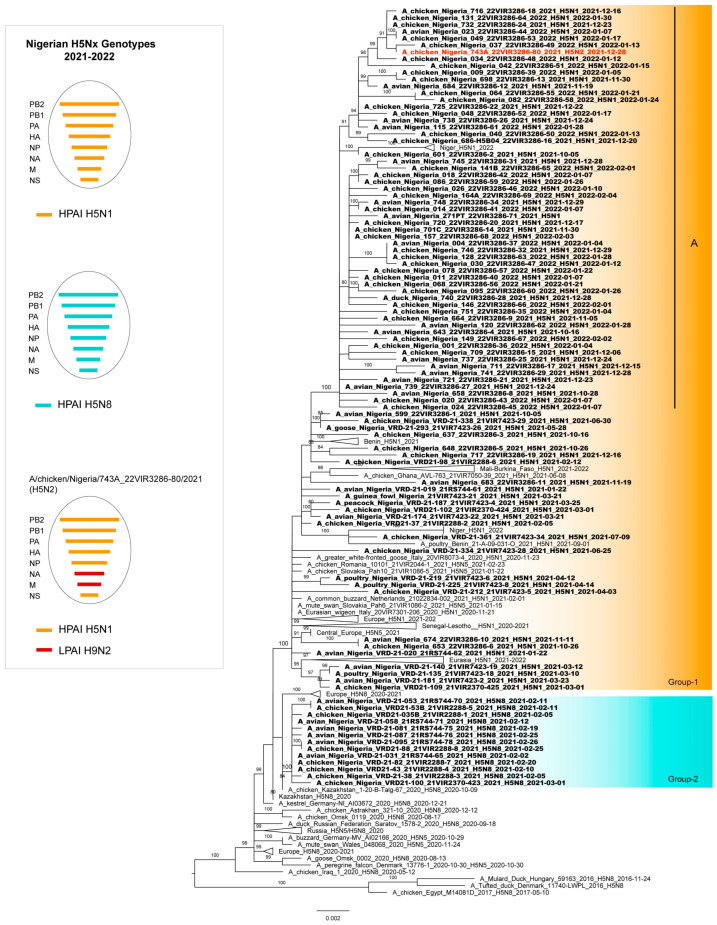
Phylogenetic tree of the nucleotide sequence of the HA gene obtained with the Maximum Likelihood method by using IQ-TREE v.1.6.12. H5Nx Nigerian viruses analysed in this study are in bold. The orange box (Group-1) includes the 2021-2022 H5N1 (black) and the reassortant H5N2 (red) Nigerian viruses. The light blue box (Group-2) includes the H5N8 Nigerian viruses. The Ultra-fast bootstrap values higher than 80% are indicated next to the nodes. The ovals next to the tree show the genetic composition of the three different genotypes (H5N1, H5N8, and H5N2) characterized in Nigeria between 2021 and 2022. Horizontal bars within each oval represent the 8 gene segments, coloured according to their origin (orange: H5N1; light blue: H5N8; and red: H9N2).

**Table 1 viruses-15-01387-t001:** Amino acid markers associated with specific phenotypic effects found in the genome of HPAI H5Nx viruses from Nigeria sequenced in the current study.

Protein	Genetic Marker	Phenotypic Effect	Virus Name	References
PB2	T105V	May be associated with human adaptation (statistical analysis)	A/chicken/Nigeria/648_22VIR3286-5/2021	[56]
	D701N	Mammalian adaptive marker	A/chicken/Nigeria/743A_22VIR3286-80/2021 (H5N2)	[41,42,43,44,45]
	K482R	Increase polymerase activity in mammalian cell line	A/avian/Nigeria/271PT_22VIR3286-71/2021	[46]
PA	D55N	Host specificity marker through statistical methods (D in avian, N in human)	A/guinea_fowl/Nigeria/VRD-21-169_21VIR7423-21/2021	[57]
	S409N	Host specificity marker through statistical methods (S in avian, N in human)	A/chicken/Nigeria/040_22VIR3286-50/2022	[57]
HA	PLREKRRKR/ GLF	HPAI cleavage site	All viruses	Not available
	S137A	Increased α2-6 sialic acid virus binding	All viruses except for A/avian/Nigeria/VRD-21-019_21RS744-61/2021	[33]
	S158N	Increased α2-6 sialic acid virus binding	All viruses	[34]
	T160A	Increased α2-6 sialic acid virus binding	In all the H5N8 strains, and in eighteen H5N1 viruses	[34,35]
NA	44-65 del in the stalk region	Enhanced virulence, marker of viral adaptation from wild bird to poultry	In 73 out of 82 H5N1 viruses	[37,38,39,40]
NP	Y52H	Confer resistance to BTN3A3 (butyrophilin subfamily 3 member A3) inhibitor	In 82 out of 83 H5N1 viruses	[47]
M1	N30D, T215A	Increased virulence in mice	All viruses	[48,49]
M2	V27I	Associated with amantadine resistance	A/avian/Nigeria/741_22VIR3286-29/2021 A/chicken/Nigeria/751_22VIR3286-35/2022	[52]
	S31N	Associated with amantadine resistance	A/chicken/Nigeria/743A_22VIR3286-80/2021 (H5N2)	[50,51]
NS1	P42S	Increase virulence in mice and could be critical to antagonize the interferon induction	All viruses	[53]
	P87S	Host specificity marker through statistical methods (S in human, P in avian)	All viruses with the exception of A/chicken/Nigeria/725_22VIR3286-22/2021 A/avian/Nigeria/115_22VIR3286-61/2022	[58]
	L103F	Increase virulence in mice	All viruses	[54]
	I106M	Increase virulence in mice	All viruses	[54]
	K217E	Decrease replicative or pathogenic potential of the virus	A/chicken/Nigeria/157_22VIR3286-68/2022	[59]
	N205S	Decreased antiviral response in host	All H5N1 viruses	[55]
NS2	T48A	Decreased antiviral response in host	All H5N1 viruses	[55]

## Data Availability

The consensus sequences of the viruses analyzed in this study were submitted to GISAID’s EpiFlu™ Database.

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
