# Peer review of "The Evolution of Highly Pathogenic Avian Influenza A (H5) in Poultry in Nigeria, 2021–2022"

_viruses, 2023, doi:10.3390/v15061387_

Round 1
Reviewer 1 Report
In this manuscript, Clement Meseko et al. reported the highly pathogenic H5Nx influenza viruses infection in Nigeria from 2021 to 2022. The globally circulating H5 viruses since 2019 have produced a huge devastation to poultry industry. The authors here detected 97 H5 viruses and characterized their genetic evolution. The study will contributed to our understanding the highly pathogenic H5 viruses infection in West Africa, however, I have several concerns that should be addressed by the authors.
1. Line 119, specific primers were used to identify the HA and NA subtype, which should be listed in the text.
2. Although, the outbreaks have been indicated by the map, I suggested the authors listed the detailed information of these outbreaks, including birds (chickens or ducks), infection numbers, subtype of the viruses, time and regions in figure 1 or in a new table. By the way, I can not open suppl table1.
3. The detailed information of the 97 should be shown in the text by a table, including virus name, host, collection date, collection sites, subtype, accession numbers.
4. Were the 97 viruses isolated from the clinical samples? If not, how many viruses were sequenced directly from the samples and how many viruses were sequenced from the isolates?
5. I strongly recommend the authors to update the phylogenetic tree of HA,and added representative viruses that detected from Asia, Europe, America to reveal the whole evolutionary trend of H5 viruses in the world.
6. The phylogenetic trees of NA and the six internal genes should be provided in the manuscript.
7. The amino acids mutations at position 30 and 215 in M1 protein have be proved to affect the pathogenicity of viruses in mammals, and should be added in the table (PMID: 19117585, PMID: 34908445 ).
8. In the discussion part, the strategies against the highly pathogenic H5 viruses in Nigeria and China should be discussed. Because China successfully avoid the large scale infection of H5 in this 2019-2023 wave by using the vaccination strategy (PMID: 36458831, PMID: 34757542).
The discussion part should be improved.
Reviewer 2 Report
An interesting account of HPAIV epidemiology in Nigeria is presented. A lot of sequencing data have been established to allow following the spread of virus within the country and to speculate about possible incursion routes. While the overall design of the study is sound, there are some conflicts as regards the validity of some of the conclusions.
1. Has there really been "extensive" HPAIV evolution and diversity in Nigeria? This reviewer wouldn't opt for this phrase based on the data shown. Compared to Europe 2020-2 and the Amercian continents after 2021 variation of viruses might at best be called "substantially" diverse.
2. Merely three genotypes have been detected, one represented only by a single virus.
3. HA variation within those genotypes seems limited and most of the variation is shown to be due to incursions from abroad.
4. This also challenges the designation of Nigeria as a "hot spot" of HPAIV evolution. Rather, from the data shown here, it seems to be a sink with very little dynamics within the country. Also, comparable surveillance data from neighboring countries is missing to allow for comparisons.
This criticism is not meaning this would not be a good paper; just the interpretation of data should be re-considered and the conclusions re-directed.
Minor suggestions:
Title: Should be re-phrased
96 - pers. comm. It is strange to see this essential and central information as a pers. comm. Shouldn't the owners of these data be included as co-authors to fully validate their findings?
202 - The genotype groups named here, how do they fit into the European nomenclature system (of which several colleagues are co-authors here)?
211 - Is it possible H5N8 has been replaced by H5N1 by greater fitness of the latter? Any indications?
233 - The bootstrap values play an important here to distinguish HA clusters. For each meaningful cluster they should be shown at an enlarged font size while the other values should be omitted completely.
248-50 - Mentioning of the glycosilation pattern and its importance: but what exactly are the changes observed with the Nigerian viruses, if any?
253 - Which viruses did not show the NA stalk deletion and how do the authors think the stalk had been completed?
365 - Why do the authors exclude the east-west movements of (infected) wild birds here?
247 - "have", not "has"
Reviewer 3 Report
In the reviewed manuscript, the prevalence of avian influenza A virus in poultry in Nigeria was investigated. The samples were collected from farms reporting the HPAI suspicions. The presence of H5N1, H5N8 and H5N2 viruses was established. Whole genome sequences of eleven H5N8, seventy-eight H5N1 and one H5N2 virus were obtained. H5N2 virus has been reliably identified as a reassortant of HPAI H5N1 and LPAI H9N2 viruses. It is shown that the viruses circulating in Nigeria are related to European and Asian viruses of the same years, and not to African viruses circulated earlier. This work expands our knowledge of the circulation of potentially dangerous viruses in Africa.
Some notes to the text are listed below:
1) Line 80-86. It would be desirable to expand this paragraph and clarify in which cases HPAI H5Nx and H9N2 reassortants were discussed, and in which cases non-pathogenic H5N2 viruses were discussed.
2) Line 175-178. It is not clear whether the entire outbreak was caused by the H5N2 virus?
Round 2
Reviewer 1 Report
The authors have addressed my concern.
Reviewer 2 Report
All my comments and suggestions have been satisfactorily answered.